# Assessment of a Weak Mode of Bacterial Adhesion by Applying an Electric Field

George Araujo, Zhaoyi Zheng, Jae Jong Oh and Jay X. Tang *

Physics Department, Brown University, Providence, RI 02912, USA; gbarbosa.a@gmail.com (G.A.); zhaoyi_zheng@alumni.brown.edu (Z.Z.); jae_jong_oh@alumni.brown.edu (J.J.O.)
* Correspondence: jay_tang@brown.edu

**Abstract:** Microbial attachment to surfaces is ubiquitous in nature. Most species of bacteria attach and adhere to surfaces via special appendages such as pili and fimbriae, the roles of which have been extensively studied. Here, we report an experiment on pilus-less mutants of *Caulobacter crescentus* weakly attached to polyethylene surface. We find that some individual cells transiently but repeatedly adhere to the surface in a stick-slip fashion in the presence of an electric field parallel to the surface. These bacteria move significantly slower than the unattached ones in the same field of view undergoing electrophoretic motion. We refer this behavior of repeated and transient attachment as "quasi-attachment". The speed of the quasi-attached bacteria exhibits large variation, frequently dropping close to zero for short intervals of time. We propose a polymeric tethering model to account for the experimental findings. This study sheds light on bacteria–surface interaction, which is significant in broader contexts such as infection and environmental control.

**Keywords:** bacterial motility; bacterial adhesion; surface adhesion; transient attachment; reversible binding; electrophoresis; galvanotaxis

## 1. Introduction

Ubiquitous in nature, microbial attachment to surfaces has been widely studied [1–3]. Adhering to human and animal tissues is often the first step towards bacterial infection [4]. Bacterial attachment is also the most crucial step for the formation of biofilms, a community of cells that form complex structures on surfaces [5,6]. Many species of bacteria are motile. They can actively explore their surrounding fluid environments. Bacteria often encounter a boundary with the aid of swimming motility and Brownian motion [7]. Motile bacteria also tend to accumulate near a boundary surface [8–10]. When within 2–3 μm from a solid surface, they may become sterically or hydrodynamically trapped and remain in proximity of the surface for significant lengths of time [8,11]. Whereas near surface accumulation and entrapment are distinct from the actual attachment, they help facilitate bacterial adhesion. Other physical effects are also known to facilitate bacterial adhesion. Shear flow, for instance, has been shown to act as a cue for surface adhesion by triggering intracellular signaling and causing *Pseudomonas aeruginosa* to transition from the planktonic to the biofilm phenotype [12].

Attachment to surfaces can be classified according to its molecular basis, as well as its strength and reversibility. Most bacterial species attach and adhere to surfaces via special appendages such as pili or fimbriae, which are flexible filaments with sticky ends. Once stuck to a solid surface, the pili can retract, pulling the cell body in contact with the surface [13]. The initial attachment by pili and their retraction is known to lead to strong and permanent attachment, the mechanism of which has been extensively studied [1]. Less is known, however, about other forms of attachment, particularly those weaker and more reversible, even though they play functional roles. For instance, weak and reversible adhesion has been shown to enhance surface colonization [14]. Sequences of attachment and detachment of binding tethers may serve as pre-play in the process of consolidating

irreversible attachment [15]. These transient attachments are also linked to optimal surface exploration as they have been proposed to maximize bacterial surface diffusivity [16].

Bacterial colonization of a surface can be undesirable in certain situations, causing various types of infections [17]. As a consequence, different techniques have been applied in order to remove bacteria from surfaces. Common examples involve flow, which may be produced by pulsating flow [18] or through passage of air-bubbles [19]. With their surface charged in the fluid environment, bacteria may be dislodged from a conducting surface, such as glass coated with indium tin oxide (ITO), by applying an electric current with the conducting surface serving as either the anode or the cathode [20–23]. One study observed that electric forces parallel to a surface are more efficient at promoting detachment than those applied in the perpendicular direction [22]. Another recent study reports strong inhibition of adhesion of *Pseudomonas aeruginosa* on the surface of ITO coated electrode by an applied direct or alternating electric current (DC or AC) [23].

We chose a Gram-negative bacterium called *Caulobacter crescentus* as a model system for the study of bacterial adhesion. Commonly found in aquatic media, including soil and drinking water [24], *C. crescentus* has a dimorphic life cycle: one newly divided cell is motile and called a swarmer. It sheds its single flagellum in 45 minutes, hence losing its motility. It then develops into a stalked cell. Under favorable conditions, the stalked cell proceeds to the next division, producing a motile swarmer cell as its offspring every two and a half hours [24,25]. The stalked cell is known to produce a holdfast capable of strongly and permanently adhering to a solid surface [26]. However, in most laboratory based adhesion studies of *C. crescentus*, the initial attachment tends to occur early in the swarmer stage. Effective attachment to surfaces for this species is aided by the presence of type IV pili and flagellar motility [27]. In fact, the swarmer attachment has been shown to trigger just in time expression of holdfast and accelerate the swarmer cell to stalked cell transition [28]. Aside from strong adhesion and permanent attachment via its holdfast, facilitated by pili [27], we focus in this work on mutants that lack pili in order to study attachment due to a different mechanism. A recent study on the tethered motion of *C. crescentus* swarmer cells of the same strain refers to the attachment without pilus as a "fluid joint", possibly formed by polymers on the cell surface [29]. Although the joint or tether the authors propose is invisible, they have indicated its location on the cell body through video analysis of the tethered motion. In another study on the surface attachment of the wild type species of this bacterium, it has been reported that the curved shape of *C. crescentus* may help the cell to colonize surfaces [30].

In this study, by applying an electric field parallel to a plastic surface coated by a charge neutral polymer, polyethylene, we perform experiments based on the initial observation that some weakly attached cells are readily detached while others remain stuck to the surface. More interestingly, a small number of tethered cells exhibit the characteristic of moving slowly along the surface, driven by the field, but never leaving the surface proximity. These particular cells move at erratic speeds. They alternate between being transiently mobile and tethered in a stick-slip fashion, which we call *quasi-attachment*. This behavior leads to the hypothesis that cell–surface polymers create weak, non-specific interactions, which can be dislodged by the applied electric force. This report focuses on this peculiar behavior and explores the mechanism by quantitative analysis of the rare occurrences of quasi-attachment for the bacterial cells in close comparison and contrast with colloidal beads under the same experimental setting.

## 2. Materials and Methods

Two strains of *Caulobacter crescentus* were used in this study: CB15 ΔPilin (provided by Yves Brun when he was at Indiana University) and SB3860 [7], which is a mutant derived from CB15 ΔPilin (kindly generated by Bert Ely from University of South Carolina, Columbia, SC, USA). Both bacterial strains lack pili, making them less likely to adhere to surfaces. SB3860 only swims forward.

*Caulobacter crescentus* was grown in peptone yeast extract (PYE) medium containing 0.2% bactopeptone, 0.1% yeast extract, 2 mM $MgSO_4$, and 0.5 mM $CaCl_2$. The bacterial growth was followed by a synchronization method [28], in order to select swarmer cells, which were the focus of the study. The culture containing primarily swarmer cells was diluted in DI water, in a 1:3 proportion of PYE to water, resulting in a relatively low ionic strength and a cell density convenient for experimental observation.

In a control experiment, a concentrated suspension of 2.0 μm diameter polystyrene beads in 1% volume fraction (Polysciences Inc., Warrington, PA, USA) was first diluted 1000× in DI water, and then to a 1:3 ratio of PYE to water mixture so that the beads were subjected to the same ionic condition as the bacteria.

A sample containing either bacteria or polystyrene beads was inserted into a capillary channel mounted on a microscope slide for observation (μ-slide I luer from Ibidi, Munich, Germany). We chose to use a channel type that was coated with polyethylene and tissue culture treated with plasma, which we found to adequately suppress the electro-osmotic effect. The channel dimensions were 50 mm in length, 5 mm in width, and 0.4 mm in height. Both ends of the channel were sealed with agarose gel (0.5% in weight of agarose in a 50 mM $Na_2SO_4$ solution), as illustrated in Figure 1. Specifically, liquid agarose mix was gently deposited at one end of the channel previously filled with the bacteria containing liquid. The slide was gently tilted for the gel mix to enter the channel by a millimeter or two. After a 2 min wait for the gel to solidify, another drop of agarose mix was added to the other end of the channel, resulting in both ends of the capillary being sealed by the agarose gel (see Figure 1 for channel visualization). The agarose gel prevents electrochemical products at the electrodes from spreading into the channel and adversely affecting the bacterial behavior [31].

a
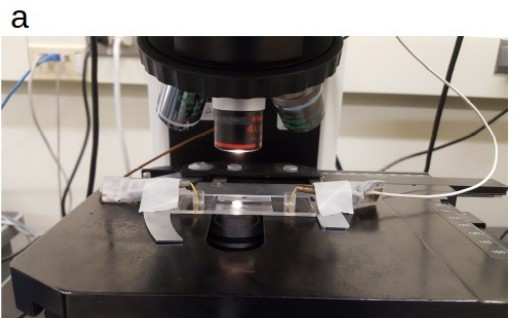

b
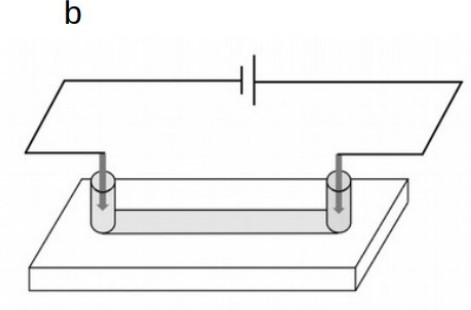

**Figure 1.** Experimental setup. (**a**) Picture showing a sample slide on a microscope stage. Two wires are connected to electrodes placed at both ends of a capillary filled with bacteria containing liquid. The microscope slide is placed so that a part of a capillary channel is imaged while an electric voltage is applied; (**b**) a simple sketch illustrating the electric circuit.

The electrodes were made of gold (99.99% purity, from Sigma-Aldrich, St. Louis, MO, USA), which is an excellent conductor. The practice of sealing the channels with agarose gel and choosing an inert metal came from the observation that the products of chemical reactions happening at the electrodes were harmful to bacteria, and a less restrictive setup had the potential to kill most bacteria within a few seconds under applied voltages over 50 V. DC voltages up to 120 V were provided by a power supply (CSI12001X, Circuit Specialists, Tempe, AZ, USA). Electric currents were measured by adding an ampmeter (DROK, Changzhou, Jiangsu, China) in series to the circuit, resulting in an I–V curve shown in the Supplementary Materials. Additional measures in order to suppress fluid flow, electro-osmotic effect, and microbubbles are described in the Supplemental Document.

Images were acquired using phase contrast under an upright microscope (Eclipse E800; Nikon, Minato City, Tokyo) with 4× or 40× objective lenses (Nikon), depending on the size of the region interested. Images were recorded with a charge-coupled device camera

(Coolsnap EZ; Photometrics, Tucson, AZ, USA). The acquisition process was controlled using Metamorph software (Molecular Devices, San Jose, CA, USA). The frame rates for video recordings were either 34 or 10 frames per second (fps).

For image processing, bacteria were tracked using ImageJ and AItracker, an artificial intelligence particle tracker based on neural networks [32]. The bacterial trajectories which were used in the velocity calculations shown were processed using a third order Savitzky–Golay filter with window length of 5. This step allows the experimental data to be smoothed, filtering some noise, but keeping essential features. The tracking of beads was done using the MTrackJ plugin in ImageJ. To smooth out the noise in the calculated velocity of the beads, we took the running average over 5 consecutive intervals for the motion of beads.

### 3. Results

#### 3.1. Types of Motion under an Electric Field

Under an electric field, four different behaviors were observed for individual cells: 1. Non-swimmers, including stalked cells and possibly some dead cells, drifted towards the positive pole, driven by the electrophoretic force caused by the electric field. 2. Cells that remained nearly immobile for some time, at most jiggling about their tethered positions due to Brownian motion. Some of these cells were seen to be spinning with respect to fixed positions before an electric field was applied, an indication of surface attachment through body or flagellum tethering. The spinning motion is driven by the cell's flagellar motor. 3. Swimmers that kept their swimming motion, either along with (rare, <10%) or against (most often, >90%) the direction of the field. 4. A small number of cells (very rare, <1%), which initially attached to a solid surface, moved along the field slowly in a slip and stick fashion. In most cases, the previously attached bacteria would either detach immediately after the field was applied, or they would never detach during the observation. The first three types of motion, which are briefly assessed in the next sub-section, are more common and better understood. The type 4 behavior, both surprising and significant, is shown separately and then contrasted with results of a control experiment using polystyrene beads. A model of weak adhesion due to polymeric tethering is proposed to account for the type 4 behavior for *C. crescentus* under an applied electric field.

We note at the onset that the four types of behaviors are not from different types of cells. Instead, other than the few exceptions of stalked and predivisional cells, the same swarmer cells in the majority manifest all types of behaviors depending on whether they are free moving in the medium or stuck on the capillary surface. The type 4 behavior is very rare, mainly because these cells are being detached by the field of just the right strength that matches their strength of attachment. If the field strength were slightly weaker, the cells would just remain stuck on the surface. If the field strength were significantly stronger than that required to sustain a stick and slip motion, the cells would just be quickly detached and join the majority as free moving or swimming cells in the medium. In other words, type 4 cells are rare transients from type 2 to type 3 (or, by extension, type 1).

#### 3.2. Alignment of Cell Trajectories under an Electric Field

In the absence of external electric field, the swarmer cells of *C. crescentus* swim in random directions, which change stochastically in the time scale of seconds (Figure 2a). Once exposed to an electric field, the effect on the motion of swimming bacteria is immediately observed, via a well-known effect called galvanotaxis [33]. The applied electric field imparts a drift speed on every bacterium, motile or not, but the effect on swimming cells is more dramatic as the field also aligns their swimming trajectories. The average direction of motion is opposite to the direction of the electric field, indicating negative net charge of the bacterial cell. The alignment in trajectory is mainly caused by orienting the cell-flagellum axis, with the cell body heading towards the positive electrode and the flagellum pushing from behind in most cases. This observation indicates a larger magnitude of electrophoretic mobility for the cell body than for the flagellum. Under stronger fields, the paths of motion

become closer to long and straight lines that are nearly anti-parallel to the field direction (Figure 2b).

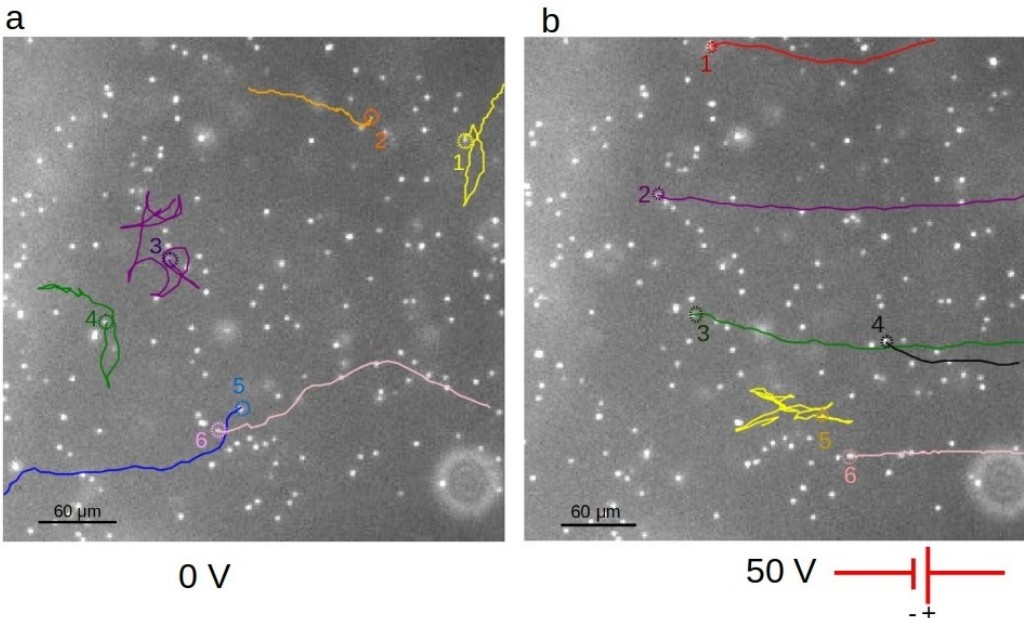

**Figure 2.** Bacteria trajectories. (**a**) Trajectories of several CB15 ΔPilin swimming cells in the absence of an electric field. Note the numerous reversals in swimming direction for cells 1, 3, and 4, due to frequent switches in the direction of flagellar motor rotation. (**b**) Under the voltage of 50 V applied, bacteria move predominantly against the applied field. The direction of the field on the figure goes from right to left. The dashed circle on each trajectory indicates the starting point of the track. The bright dots are images of bacteria that were either stuck to the surface or moving near the surface. For clarity, only six cells are highlighted on each figure. The trajectories were traced over 12 s. Some cells moved beyond the borders of the imaging area during the observation period. The images were taken using a 4× objective lens. The expansion of the bright dots due to optical diffraction makes them appear bigger than the actual size of the bacteria.

It is interesting to note that some cells are observed to swim in the direction of the field, i.e., towards the negative pole, typically at much slower speeds. In these cases, the swimming direction is parallel to the electric field. We know from previous studies that, when the flagellar motor of *C. crescentus* rotates counter-clockwise, the flagellum pulls the cell body backward [7,34]. This backward swimming motion appears to have overcome the electrophoretic effect, resulting in net slow motion downstream with respect to the electric field. In the absence of an electric field, *C. crescentus* switches the direction of swimming within a few seconds [35] (unless genetically modified to prevent the switching behavior, such as in the SB 3860 strain, which swims solely forward). We indeed observed switches under the applied field, but an arbitrarily large change in the direction of motion as the result of a flick [36] was suppressed. Even with changes in the direction of motion, the trajectories remain aligned with the electric field. One example is shown as trajectory 5 in Figure 2b. In short, immediate and large changes in trajectories of swimming bacteria under an applied electric field can be accounted for by taking into consideration flagellar motor driven propulsion, electrophoretic mobility, and, most notably, alignment due to electrophoretic forces acting on both the cell body and the flagellum.

### 3.3. Speed Variation of Weakly Attached Cells Moving under the Field

Unattached cells, whether swimming against the direction of the electric field or solely moving electrophoretically, travel faster as the field strength is increased. In contrast, the slowly dragging cells behave differently. These cells transit from short moments of limited mobility to moments of negligible mobility. We call the weak interaction of these

cells with the plastic surface "quasi-attachment". Figure 3a,b highlights an example of this behavior (labeled as cell 2), in contrast to cells that are either permanently attached to the surface (cell 1), or drifting electrophoretically (cell 3). Movie S1 in the Supplemental Materials shows a video from which the images were taken. While cells freed from the surface attachment quickly move away from their initial positions, the motion of a quasi-attached cell is much slower, as indicated in the images, with its displacement and speed plotted in Figure 4.

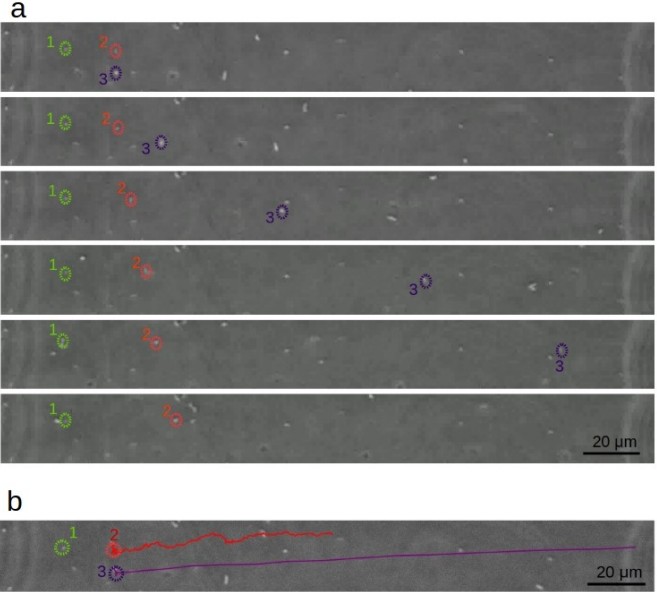

**Figure 3.** Types of bacterial motion under direct current (DC) electric field. (**a**) Cell 1 is attached to the surface and hardly moves. Cell 2 moves slowly under the field (quasi-attached), making frequent changes in speed. Cell 3 was initially attached to the surface, but it detached quickly after the electric field was applied and then moved fast and freely along its path. The images in the sequence are 2 s apart. The voltage applied was 100 V; (**b**) tracing the trajectories as illustrated in Figure (**a**) over 31 s. Note that the time interval traced is longer than what is shown in the time lapse sequence in (**a**). These images were taken using a 40× objective.

During the short intervals when a quasi-attached cell was moving, its velocity component along the field axis remained highly variable, as is shown in Figure 4b. It is possible that this noisy speed–time curve was also due to very brief moments of attachment, but we could not confirm that due to limitations in image acquisition. In this study, all speed values were averaged over about one tenth of a second, which was set by both pixelation accuracy and the frame rate of the images taken.

One surprising behavior of these bacteria in draggy, stick, and slip type of motion is that no obvious correlation is noted between the intensity of the applied field and the average moving speed of the cells (in the Discussion section, we elaborate on this peculiar property). Figure 5 shows results of six quasi-attached cells (strains CB15 Δ*Pilin* and SB 3860) of averaged velocities along the electric field axis under different electric fields. There are large variations in each cell's moving velocity over time, as indicated by the large bars representing the standard deviations. The average velocity also varies among the six cells, but there is no obvious correlation between the average velocity and either the voltage applied or the strain of the cells tested. The velocity of the dragged motion of two cells of the same strain in the same experiment, denoted as SB 100V (i) and SB 100V (ii), also differed by a factor of 2, albeit each manifesting large variations over time. Note that the trajectories of these cells are aligned with the field axis. Thus, the velocity component along the field axis properly accounts for the average drift velocity of these transiently attached cells.

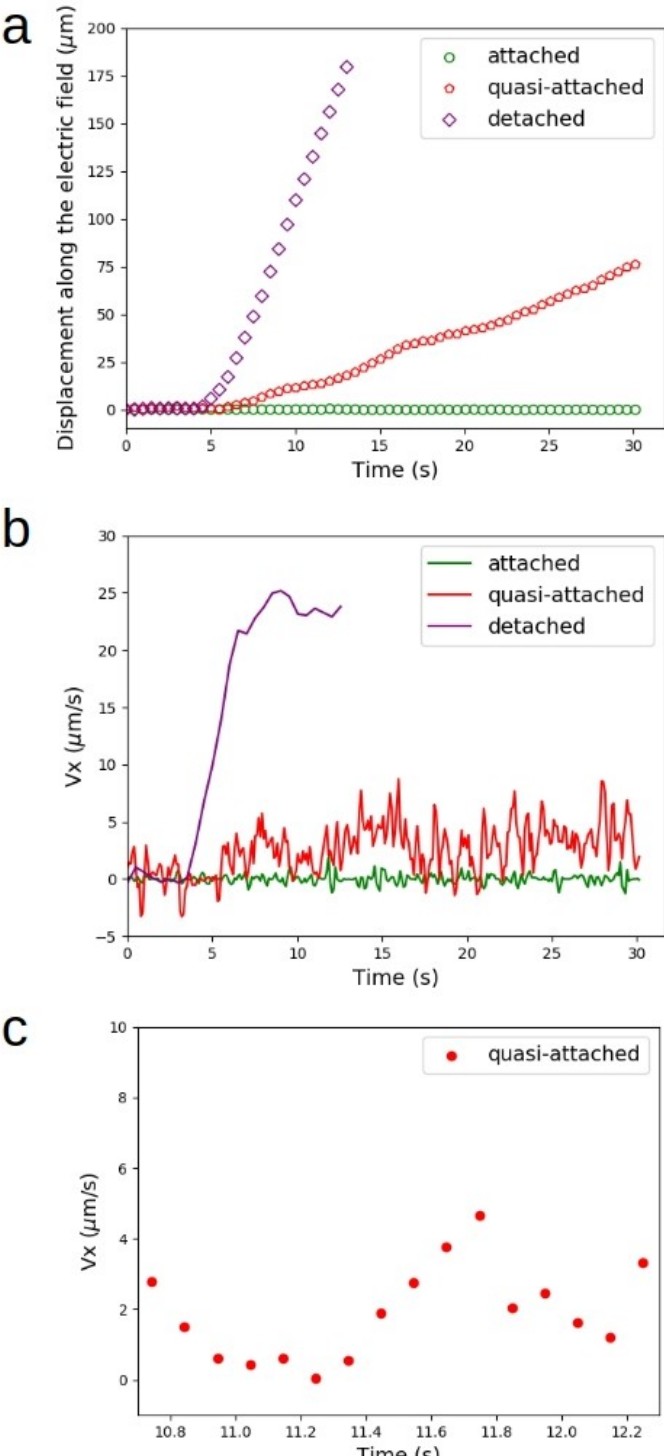

**Figure 4.** Displacement and velocity along the field. (**a**) Displacement of the highlighted cells along the axis of the electric field. Experimental data are shown for points separated by 0.5 s, which is five times the time interval for data acquisition. The voltage applied was increased from 0 V to 100 V within seconds, and then kept at its maximum value. (**b**) Velocity along the field as a function of time for the three highlighted cells shown in Figure 3. (**c**) Zoomed in plot of velocity values lasting for approximately 1.5 s of the data in the plot shown in **b** above. Within this short time interval, we see in more detail the variation of velocity along the field. From 10.9–11.3 s, for instance, the velocity was close to zero, indicating a moment of transient attachment.

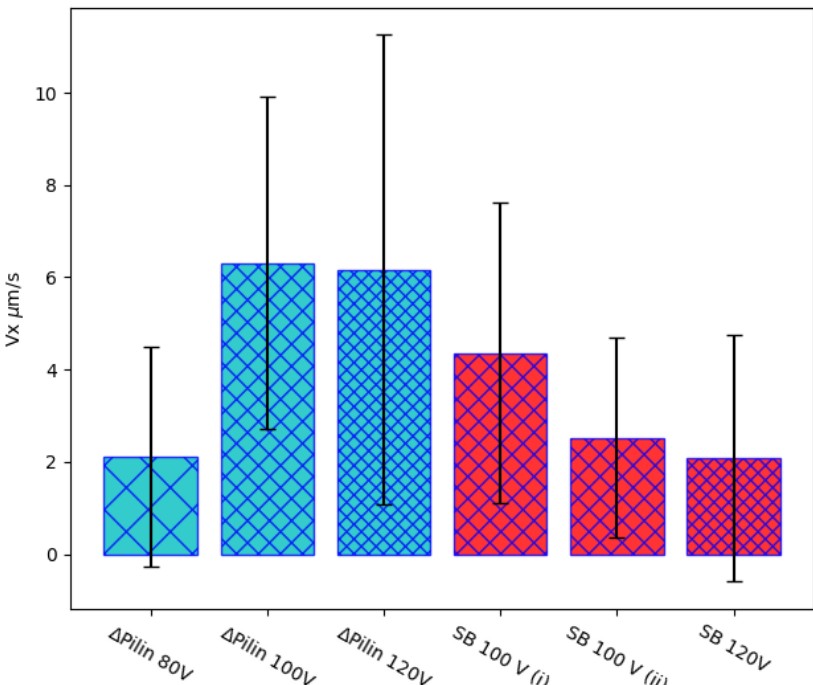

**Figure 5.** Average velocity component along the axis of the electric field for six cells in dragged motion from two different strains and under three different values of applied voltages. SB 100 V (i) and SB 100 V (ii) represent two cells tracked from the same video, showing a cell to cell variation by a ratio of 2 to 1. Representing standard deviations, the large error bars indicate notable velocity variations over time for each individual cell. This is a characteristic behavior of quasi-attached cells in stick-slip motion.

*3.4. Control Experiment of Polystyrene Beads Transiently Attached to the Same Plastic Surface under an Electric Field*

In order to discern to what extent the quasi-attachment of bacteria on the plastic surface might be caused by non-specific physical forces, such as those attributable to the classic Derjaguin–Landau–Verwey–Overbeek (DLVO) theory [7,37], we performed a control experiment using colloidal beads under the same setup. We used polystyrene beads comparable in size to bacteria, diluted in the same solution as used for the bacteria. We found that a much lower voltage was required to dislodge the small number of beads that occasionally attached to the plastic surface. The range of voltage was 15–25 V, as opposed to 80–120 V for the bacteria. Such a difference in strength of attachment was notable. Nevertheless, we found a small number of them in both cases transiently attached to the plastic surface, thereby allowing for a close comparison.

The beads near the plastic surface behaved very differently from the bacteria, motile or not. Most beads drifted electrophoretically under the applied field. In rare instances, some beads became transiently immobile for variable intervals of time on the order of seconds. After freeing itself from transient attachment, such a bead usually resumed its electrophoretic motion of the same velocity as before, a velocity comparable to the majority of beads that never became attached. Figure 6a shows a trajectory of a transiently attached bead in comparison with a freely drifting bead under an applied voltage of 15 V (see also Movie S2 in the Supplemental Materials). The velocity profiles of these two beads are plotted in Figure 6b. Two other examples are shown in separate experiments under slightly higher applied voltages (Figure 6c,d). All of these examples show a very different behavior from the quasi-attachment of bacteria. Whereas the latter displayed dragged motion of nearly continuous attachment, the transient attachment of a bead was binary: it appeared either totally attached or detached, and there was no lingering effect on the bead's drift speed once it was detached.

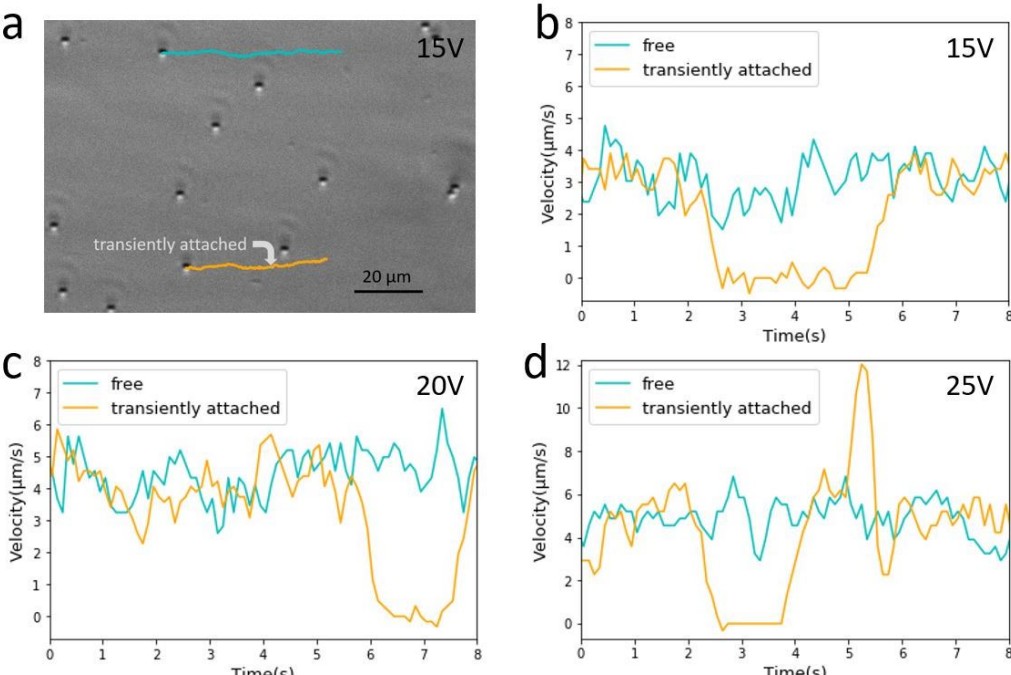

**Figure 6.** Transient attachment of colloidal beads in a control experiment. (**a**) First frame of a video capturing electrophoretic motion of beads under 15 V applied voltage. Superimposed are two trajectories, one of a freely drifting bead (cyan), and the other also drifting (yellow) except for a 3-s pause at a spot indicated by an arrow. (**b**) Velocity versus time plots of two beads from their trajectories indicated in (**a**,**c**,**d**). Examples of similar beads motion observed in two other experiments under slightly higher applied voltage values of 20 V and 25 V, respectively. The images were taken using a 40× phase contrast objective lens. The positions of beads were captured in 0.1 s intervals, but the velocity values were 5-point running averages in order to reduce noise in the plots.

## 4. Discussion

### 4.1. Charge Distribution along the Bacterial Cell Is Primarily Responsible for Trajectory Alignment

The cause for the alignment of bacterial trajectories under an electric field is physical in nature: there is a difference in electrophoretic mobility between the cell body and the flagellar filament [33]. The typical surface area of the cell body is in the range of 2–5 $\mu m^2$ [38], which is around two orders of magnitude bigger than the surface area of the flagellum, whose averaged contour length is a few $\mu m$ but whose cross-sectional diameter is only about 14 nm [39]. The hydrodynamic drag on the cell body is also larger than on the much thinner flagellum. Based on our observation that motile cells swim towards the anode (positive electrode) faster than the drift of non-motile cells, we know that the cell body usually leads the flagellum. In this orientation, the bacteria are both driven by the field and pushed by the flagellum towards the electrode of higher voltage. The experimental observation also indicates that the ratio of the drag coefficient is much less than that of the effective charges between the two. Thus, the imposed cell body–flagellum orientation dictates the kinematics of the bacterial trajectories. Under strong fields (with applied voltages above 40 V over a 5 cm distance in our experiments), cell trajectories do not deviate much from straight lines even for cases when we observe switches in the direction of motion. This result is due to the alignment of the cell body by the electric field, which may also suppress the flagellar flicks [34]. For *C. crescentus*, with its flagellum of right handed helical structure, a flick occurs as its flagellar motor switches rotation from counter-clockwise to clockwise, causing the bacterium to change from backward to forward motion. A flick often causes the bacterium to reorient its cell body orientation, causing a large change in swimming direction in the absence of an applied field. The presence of a strong electric field keeps the cell body aligned, thus suppressing the randomizing changes

in swimming direction caused by the flicks, regardless of whether or not the frequency of switches in motor rotation directions is affected.

### 4.2. Comparison in Forces of Detachment by Several Techniques

Applying an electric field to detach cells from a surface offers a unique range of forces as compared with other techniques. In our experiment, the typical force applied is in the range of 0.1–0.5 pN (estimated in Supplemental Document). Such a strength of force is 1–2 orders of magnitude stronger than that achieved by a common micro-fluidic approach [40]. However, it is 1–2 orders of magnitude smaller than the force that can be exerted by laser tweezers (1–100 pN) [41,42], and several orders of magnitude smaller than forces applied by atomic force microscopy (AFM) (orders of pN-nN) [43] or micromanipulation techniques (up to µN) [26]. The strength of bacterial adhesion spans such a wide range of magnitudes, which has been demonstrated by experiments applying all these techniques. Interestingly, the present study shows that forces in the sub-piconewton range are required to detach the swarmer cells of two pilus-less strains of *C. crescentus* that weakly and non-specifically attach to a plastic surface. Such a weak interaction might be a significant prelude to much stronger and permanent attachment to surfaces mediated by pili and/or other appendages. However, measurement of the tethering force is difficult, perhaps due to a lack of appropriate tools suitable in this particular range of magnitude. Although relatively weak attachment, what we observed and measured in this study may be among the interactions that are stronger than those that can be conveniently washed off by moderate shear flow, limiting the effectiveness of conventional micro-fluidic devices (see a comparison between the electric force and shear force in Supplemental Materials). Other studies on the same pilus-less strain of the same bacterial species have shown that the interaction is nevertheless strong enough to tether the cells to a solid surface, lasting minutes and withstanding the rolling motion driven by its active flagellum [29,44]. The method of applying an electric field as a force probe that we show in this study may prove useful for future studies on a variety of microbes in the context of their attachment and adhesion to surfaces, including those under environmentally or medically significant settings.

### 4.3. The Relevance of DLVO Theory to Weak and Transient Attachment

There is an extensive literature discussing the relevance of electric double layer interaction with bacterial adhesion to surfaces, as reviewed in [45]. One study explored the role of cell surface lipopolysaccharides in *E. coli* adhesion to the surfaces of quartz beads or coverslips [46]. Using packed bed columns and a radial stagnation point flow setup, the study investigated the transient interaction of K12 strains expressing several lengths of polysaccharrides under different ionic conditions with silanized surfaces. The authors suggest that there is a combined influence of electrostatic interactions, in terms of the classic DLVO theory, LPS-associated chemical interactions, and hydrodynamics of the deposition system. Another example is a more recent study that explores electrokinetic forces on bacterial deposition and transport in porous media [47]. It includes an estimate of the DLVO force of colloidal interactions, as well as calculation of both electro-osmotic and electrophoretic forces.

The DLVO energy is given by adding the van der Waals and electrostatic contributions to the particle-surface interaction. The derivative of the total energy with respect to the particle–surface distance yields the interaction force. Using typical values to represent these quantities [7] (i.e., for glass surface in this reference study), the DLVO force is estimated to be around a fraction of a pN when the system is in the vicinity of the second minimum of energy (noted from a plot in the published Supplemental Materials of the said reference). This estimated force coincides with the range of the force provided by the electric field, supporting the idea that, while dragged along by an applied electric field, bacterial cells, like colloidal beads, may become weakly attracted to the solid surface. Aided by incessant Brownian motion, they may occasionally bump into the surface and become transiently

attached to it. We also note in the work by Shan et al. [47], a larger DLVO force of several nN is estimated between the bacteria and the glass surface, suggesting a stronger role of this physical effect on bacterial attachment to surfaces.

However, the strikingly different behavior between the transient attachment of colloidal beads and quasi-attachment of the bacteria on the same plastic surface points to the necessity of an additional, more specific mechanism than DLVO, which applies to both but could not account for their qualitatively different behaviors. Thus, the ensuing discussion examines other hypotheses, and, followed by a polymer tether model, we propose to account for the stick and slip type bacterial attachment.

### 4.4. Assessment of Tentative Hypotheses

One simplistic hypothesis is that these slow moving cells are just cells that are swimming towards the negative pole, i.e., in the field direction and opposite to the direction of the electrophoretic driving force. In this scenario, they would be moving slowly because they would be "fighting" against the electric force. However, the slow, irregular motions are seen over dozens of seconds, whereas we know from an earlier study that the characteristic time for switching the direction of motor rotation rarely goes beyond a few seconds [35] (for $\Delta Pilin$ cells; SB3860 only swims forward). It is very unlikely based on this picture that a $\Delta Pilin$ cell would be swimming over 30 s in the same direction along the electric field. In fact, we observed multiple examples of slow, dragged motion for $\Delta Pilin$ cells, thereby ruling out this model. On a separate line of reasoning, this hypothesis can also be ruled out based on the fact that we observed these slow and dragged motions at several applied voltages. It would be rather hard to achieve the nearly exact cancellation between the two opposite velocities in all these experiments since the electrophoretic velocity is proportional to the applied voltage, whereas the swimming velocity would remain constant.

A second hypothesis is that the no-slip boundary condition for fluid flow might account for the transient, slow, and dragged motion. In other words, the observed slow motion could just be reflecting the extremely slow-flowing fluid containing non-motile cells very close to the surface. This picture is inconsistent with results from an earlier study measuring the swimming speed of *C. crescentus* as a function of its distance from glass surface using total internal reflection fluorescence (TIRF) microscopy. In that study, the cells were found to swim relatively freely as close as on the order of ten nanometers from the surface, with their typical swimming speeds ∼40 µm/s [7]. The qualitatively different type of motion and the much slower and highly variable motion of the subset of cells identified in this study point us to a totally different mechanism.

Another possibility is that the flagellar filament frequently scratches the solid boundary, causing too large a drag to allow fast motion of those cells touching the surface. Given that a typical flagellar rotation rate is in the range of 250–375 Hertz [48], the flagellum could be interacting with the surface on the order of hundreds of encounters per second at some rough spots, imposing a strong drag on the bacterium. This interaction of flagellum and surface could also be a cause for the quasi-attachment. A potential method to test this hypothesis would be directly visualizing flagellum. The method would require labeling the flagellar filament while not severely affecting the flagellated motion. One might then be able to directly visualize the behavior of the labelled bacteria when they display quasi-attachments under applied electric field. This approach has been attempted previously, using genetically modified strains of *C. crescentus* so that its flagellum could be labelled. Unfortunately, the labeling resulted in much reduced motility and only in rare cases did short video segments show a motile cell with a faintly labelled filament [29]. Once a better labeling technique is developed, applying it to the experiments under an electric field as described in this report could help to clarify to what extent the flagellar filament affects the quasi-attachment.

### 4.5. The Proposed Mechanism

We propose that cell surface polymers, such as long and flexible polysaccharrides richly expressed on the bacterial surface, account for what we refer to as "quasi-attachment" of pilus-less bacteria to a solid surface. This attachment is similar to the "weak rolling mode of surface adhesion" for *E. coli* [14], yet the molecular origin is different, as the latter is known to bind to mannosylated surfaces via the adhesive protein FimH. The molecular mechanism we envision is akin to what has been proposed in a recent report [15], albeit on a different species of bacteria.

It has been proposed [29] in a previous report based on experiments on the same *C. crescentus* strain as used in this study that a "fluid joint" made up of polysaccharides that cover the cell wall could be responsible for the interaction with the surface (Figure 7). The tether region seems to be located near the flagellar pole in most cases, but occasionally also near the center of the body [29]. Given that the nature of this cell–surface attraction is proposed to be caused by polymeric links between the bacterial cell wall and the surface, we suggest to refer this interaction as *body-tethered* [44], via *polymeric tethers* [46,49]. The reason for this proposed change in terminology is that "fluid joint" implies no molecular link, but, instead, the action of a thin layer of confined fluid, which is confusing. Instead, we suggest explicitly in a simple sketch here to convey our proposed mechanism (Figure 7). We envision multiple molecular contacts, each weak, transient, and non-specific. Collectively, they result in attachment that can sustain shear flow but allow a stick-slip type of motion at a certain strength of an applied electric field.

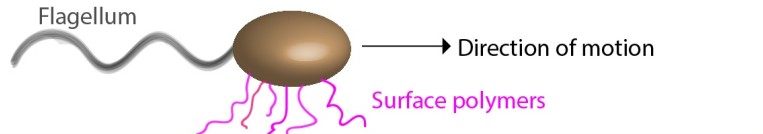

**Figure 7.** Illustration of quasi-attachment of a pilus-less bacterium to a solid surface through polymers out of the cell surface, referred to as "surface polymers". Our proposed model is that the individually transient but repeated binding of these polymers to the surface gives rise to the stick-slip type attachment.

Our proposed molecular mechanism accounts for the key observation with respect to quasi-attachment. A single, non-specific molecular interaction would likely not be strong enough to withstand the applied field. Plus, if tethered by a single bond, its breakage would immediately set the cell free, leaving a very low probability of repeated attachment. Although unable to visualize these polymers, one can envision multiple tethers with several transient contacts, which best account for the large variation of the drag speed, while the cell remains continually attached. Additionally, since *C. crescentus* has a curved shape, as the cell is frequently going through moments of attachment and detachment, the region of cell–surface contact may vary significantly. The strength of interaction, including that attributable to the DLVO theory, is expected to vary. The bonding formed between different areas of the cell and the plastic surface could lead to variations in the speed of motion of a crescent-shaped cell even if there is a constant force driving it to move parallel to the surface. Taken together, these considerations may account for the noisy velocity profiles we measured.

## 5. Conclusions

We report on the observation of frequent and transient attachments and detachments of pilus-less mutants of *C. crescentus* from a solid surface. The events were identified by applying an electric field parallel to the surface. These transient attachments and the dragged motion as the cells are repeatedly detached manifest a complex interaction between the bacteria and the solid surface. Our findings point to a weak mode of bacteria–surface interaction, which we refer to as "quasi-attachment". The physical origin of the interaction

may be caused by multiple "polymeric tethers," with individually transient but repeatable interactions. The quasi-attachment may be complemented by the non-specific cell body–surface interaction attributable to the classical DLVO theory, but we found qualitatively different behavior from bead-surface interactions lacking polymeric tethers. The model we propose based on the analysis of the "quasi-attachment" events offers mechanistic insights, which can guide further experiments on bacterial species expressing different surface polymers and/or on different types of surfaces. Therefore, the study may lead to applications involving bacterial attachment to or removal from surfaces.

**Supplementary Materials:** The following are available online at https://www.mdpi.com/article/10.3390/applmicrobiol1020019/s1.

**Author Contributions:** G.A. and J.X.T. designed the project. G.A., Z.Z., and J.J.O. performed the experiments and analysis. G.A. and J.X.T. wrote the paper. All authors have read and agreed to the published version of the manuscript.

**Funding:** This work was funded by the National Science Foundation [DMR 1505878]. We also acknowledge financial support from Capes Foundation (Brazil) (G.A.).

**Institutional Review Board Statement:** Not applicable.

**Informed Consent Statement:** Not applicable.

**Data Availability Statement:** Data is contained within the article.

**Acknowledgments:** We thank Weijie Chen for valuable suggestions. We are grateful to the AItracker (www.aitracker.net (accessed on 15 June 2021)) team for processing part of the image tracking data.

**Conflicts of Interest:** The authors declare no conflict of interest.

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
