# Peer review of "Assessment of a Weak Mode of Bacterial Adhesion by Applying an Electric Field"

_2673-8007, doi:10.3390/applmicrobiol1020019_

Round 1
Reviewer 1 Report
This paper presents an interesting topic. It is well-drafted and scientifically sound. The evaluations of results are sufficient, clear, and contain sufficient information. Most of the Figures are also clear and demonstrative. However, there are some issues to fix:
Line 109: Figure?? please provide the correct number.
A parameter that the authors didn't consider is related to the roughness of the surface. How is it? Do all the surfaces present the same morphological and topographical features (for example Ra, Rz)?
I just suggest incorporating more references in the discussion section to increase the validation of your results.
How many bacterial cells have been observed for each group? I can not find this information in the manuscript. And I think it's really important. Furthermore, the histogram reported in Figure 5 doesn't report statistical analysis and it seems there are no differences between the group. Please provide a statistical analysis )and report in the test the section referred to statistical analysis) otherwise, the graph doesn't have any sense.
Author Response
We thank the expert reviewer for the positive assessment of this work. Here are brief answers to a few pointed raised:
1. Line 109: Figure?? please provide the correct number. Answer: This is corrected. It refers to Figure 1.
2. A parameter that the authors didn't consider is related to the roughness of the surface. How is it? Do all the surfaces present the same morphological and topographical features (for example Ra, Rz)?
Answer: Our study did not explore this issue, as we relied on using commercial channels purchased from a German company. Upon optical microscopy inspection, the channel surface appears clean and smooth, much like that of pre-cleaned glass coverslit from Fisher Sci. However, we have not been able to use more proper techniques to further assess the surface features of the device used, and the company is cagey about the polymeric coating of the type of the device we purchased. It did suppress electroosmotic effect well, which we found crucial to avoid that additional complication on top of other types of motion we report in the manuscript.
3. I just suggest incorporating more references in the discussion section to increase the validation of your results.
ANSWER: We added several refs in the discussion (23, 47 & 49). In particular, we noted an extensive literature on connections between DLVO theory and bacterial adhesion. Thus, we modified the subsection in the discussion in order to convey that connection a bit better.
4. How many bacterial cells have been observed for each group? I can not find this information in the manuscript. And I think it's really important. Furthermore, the histogram reported in Figure 5 doesn't report statistical analysis and it seems there are no differences between the group. Please provide a statistical analysis )and report in the test the section referred to statistical analysis) otherwise, the graph doesn't have any sense.
ANSWER: Under "Types of motion under electric field" in the "Results" section, we report our observation in terms of how numerous or rare each type of motion was observed. Perhaps we should have counted and noted numbers observed for each type. However, since the numbers varied by orders of magnitude among the 4 types of motion, we deemed it sufficient to tell readers which types occur more frequently, with either thousands or hundreds of cells noted simply by moving the field of view under the microscope, which types of motion were rare and required careful attention to identify. Figure 5 plots fully analyzed cells of the rarest type of motion. We noted in the text that these were the only 6 cells we fully analyzed over long periods of time (>30 sec). The error bars indicate the standard deviation of each cell's speed over its recorded duration of motion (over >30 sec). We regret not having to analyze more cases, but the students completed their studies and left. The subsequent Corvid-19 pandemic ended the experimental study, making it impractical to collect more data in the foreseeable future.
Reviewer 2 Report
The manuscript is well written and argued. It presents an interesting concept that has been mentioned in literature, yet not extensively studied.
There some minor structural issues. Materials and Methods should only mention experimental set-up and not observations/results. I mention some indicative ones below and would suggest the authors carefully proof-read their manuscript again:
- Lines 118-122 would be better suited for the results or discussion section.
- Lines 129-132 would also be better suited for the results or discussion section
- Lines 138-141 would also be better suited for the results or discussion section
Line 108: “…in Figure ??.”, question marks should be replaced with a no. here (probably no. 1)
It would be interesting to get the authors’ opinion on how the electric field would affect the adhesion of gram-positive bacteria.
Author Response
Please note our brief, point-by-point, response below:
The manuscript is well written and argued. It presents an interesting concept that has been mentioned in literature, yet not extensively studied.
ANSWER: We thank the experienced reviewer for this positive assessment of the work.
There some minor structural issues. Materials and Methods should only mention experimental set-up and not observations/results. I mention some indicative ones below and would suggest the authors carefully proof-read their manuscript again:
- Lines 118-122 would be better suited for the results or discussion section.
- Lines 129-132 would also be better suited for the results or discussion section
- Lines 138-141 would also be better suited for the results or discussion section
ANSWER: Indeed! We have moved all these observations into supplemental materials, under a subsection of additional results. To us, they were important efforts to ensure that our experiments were performed properly. The information isn't crucial for readers to understand the key findings of the report.
Line 108: “…in Figure ??.”, question marks should be replaced with a no. here (probably no. 1)
ANSWER: This was a latex bug, and has been corrected.
It would be interesting to get the authors’ opinion on how the electric field would affect the adhesion of gram-positive bacteria.
ANSWER: Frankly, we did not perform this study on any gram positive bacteria to gain insight. We did perform similar experiments on a pilus deletion strain of Pseudomonas aeruginosa, another gram negative bacterium, yielding similar observations. However, that experiment was too limited for us to include in this report. We hope this report will motivate more future studies on the intriguing mode of transient bacterial attachment/adhesion, using applied electric field as a useful tool.
Round 2
Reviewer 1 Report
The authors provided part of the info requested. I would suggest other pre-test techniques to characterize the samples because this part is missing but this depends on the authors and on the availability of the company.
Please, have another check about English and style.
Author Response
We moved description of some pre-test techniques and sample characterization to supplemental materials. It now stands therein as part 1 with an explicit subtitle. This was in response to a valid criticism of the other reviewer in that we should refrain from describing results, even purely observational, in the Materials and Methods (MM) section. Whereas we felt those technical details important to covey, keeping them in the main text might divert readers from the key findings of the report. We added a sentence in MM, drawing interested readers to the materials now under the Supplemental document.
We went over the entire manuscript and made several minor linguistic and style corrections. Apologies for not finding a practical way to underline them using the Overleaf latex template.